# Computational Assessment of Spectral Heterogeneity within Fresh Glioblastoma Tissue Using Raman Spectroscopy and Machine Learning Algorithms

**DOI:** 10.3390/molecules29050979

**Published:** 2024-02-23

**Authors:** Karoline Klein, Gilbert Georg Klamminger, Laurent Mombaerts, Finn Jelke, Isabel Fernandes Arroteia, Rédouane Slimani, Giulia Mirizzi, Andreas Husch, Katrin B. M. Frauenknecht, Michel Mittelbronn, Frank Hertel, Felix B. Kleine Borgmann

**Affiliations:** 1Faculty of Medicine, Saarland University (USAAR), 66424 Homburg, Germany; 2National Center of Neurosurgery, Centre Hospitalier de Luxembourg (CHL), 1210 Luxembourg, Luxembourgisfernan@outlook.com (I.F.A.); andreas.husch@uni.lu (A.H.); 3Department of General and Special Pathology, Saarland University (USAAR), 66424 Homburg, Germany; 4Department of General and Special Pathology, Saarland University Medical Center (UKS), 66424 Homburg, Germany; 5National Center of Pathology (NCP), Laboratoire National de Santé (LNS), 3555 Dudelange, Luxembourg; 6Luxembourg Centre for Systems Biomedicine (LCSB), University of Luxembourg (UL), 4362 Esch-sur-Alzette, Luxembourg; 7Doctoral School in Science and Engineering (DSSE), University of Luxembourg (UL), 4362 Esch-sur-Alzette, Luxembourg; 8Luxembourg Center of Neuropathology (LCNP), 3555 Dudelange, Luxembourg; 9Department of Cancer Research (DoCR), Luxembourg Institute of Health (LIH), 1210 Luxembourg, Luxembourg; 10Department of Life Sciences and Medicine (DLSM), University of Luxembourg, 4365 Esch-sur-Alzette, Luxembourg; 11Faculty of Science, Technology and Medicine (FSTM), University of Luxembourg, 4365 Esch-sur-Alzette, Luxembourg; 12Hôpitaux Robert Schuman, 1130 Luxembourg, Luxembourg

**Keywords:** Raman spectroscopy, vibrational spectroscopy, glioblastoma, brain tumor, heterogeneity, machine learning, unsupervised learning

## Abstract

Understanding and classifying inherent tumor heterogeneity is a multimodal approach, which can be undertaken at the genetic, biochemical, or morphological level, among others. Optical spectral methods such as Raman spectroscopy aim at rapid and non-destructive tissue analysis, where each spectrum generated reflects the individual molecular composition of an examined spot within a (heterogenous) tissue sample. Using a combination of supervised and unsupervised machine learning methods as well as a solid database of Raman spectra of native glioblastoma samples, we succeed not only in distinguishing explicit tumor areas—vital tumor tissue and necrotic tumor tissue can correctly be predicted with an accuracy of 76%—but also in determining and classifying different spectral entities within the histomorphologically distinct class of vital tumor tissue. Measurements of non-pathological, autoptic brain tissue hereby serve as a healthy control since their respective spectroscopic properties form an individual and reproducible cluster within the spectral heterogeneity of a vital tumor sample. The demonstrated decipherment of a spectral glioblastoma heterogeneity will be valuable, especially in the field of spectroscopically guided surgery to delineate tumor margins and to assist resection control.

## 1. Introduction

With the aim of providing an unbiased approach to brain tumor diagnosis, Raman spectroscopy (RS) has been progressively developed and advanced in recent years to potentially add to the ever-expanding diagnostic toolbox for the detection and diagnosis of neuro-oncological lesions alongside existing diagnostic methods to date, namely radiological imaging, histomorphology, immunohistochemistry, genetic and epigenetic analysis [1,2,3]. As a vibrational spectroscopic technique, RS allows us to detect changes in the virtual vibrational level of the molecule or tissue of interest—the interaction of light and matter results in the emission of photons of different frequency and energy (inelastic scattering, Raman scattering). Monitoring these inelastically scattered photons enables the subsequent generation of an individual molecular fingerprint of the underlying tissue. Due to its ability of real-time, label-free, and non-destructive tissue identification, current applications of RS range from intra-/perioperative use in neurosurgery to the employment of this method in diagnostic pathology [4]. Based solely on spectral tissue properties, fast and label-free discrimination between healthy dura mater and meningioma is feasible [5]. Furthermore, Hollon et al. successfully employed RS combined with convolutional neural networks to generate virtual histological images based on inherent spectroscopic tissue features, enabling differentiation between a vast amount of different brain tumor types (gliomas, meningiomas, metastasis), and the team of Zhou et al. utilized molecular vibrational fingerprints of particular lipids and proteins to determine the grade of gliomas [6,7]. For intraoperative use, fast measurements of single spots would need to provide sufficient information about the nature of the underlying tissue in order to be useful for resection control. In case of heterogenous and infiltrating tumors, this includes the infiltrated residual tissue itself, vital tumor cells and necrotic areas. However, spectral classification typically relies on parallel identification by other means, such as microscopy, as the different components are not identifiable with a naked eye. Thus, establishing spectral subclasses within heterogenous tumors has been difficult when excluding processed specimens [8,9,10]. Here, we used independent and complementary computational methods to decipher the spectral heterogeneity within glioblastoma tissue fragments and we identified several spectral entities.

We spectroscopically measured 43 glioblastoma cases as well as gray and white matter from an autoptic brain as a healthy control. While different components of glioblastoma were present in most analyzed fragments, some almost exclusively contained necrosis, which prompted us to reduce the spectral complexity in the entire data set using random forest classification (id est, the combined output of a construction of multiple decision trees) as a supervised machine learning approach. After transferring and mapping the complexity of acquired spectroscopic features in non-necrotic tumor tissue to a lower dimensional space, we then employed the unsupervised dimensional reduction technique UMAP (Uniform Manifold Approximation and Projection). Combined with the unsupervised machine learning algorithm k-means clustering, where data are split into distinct clusters (groups) of similarity based on vector quantization, we identified several spectral subclasses. By making a comparison with autopsy tissue, we were further able to identify two of these classes as gray and white matter. The possibility to correctly classify individual measurements on single spots opens avenues to spectroscopy-assisted resection control, which is a major challenge in this field [11].

## 2. Results

In order to assess data quality, we first performed a hierarchical clustering, which did not reveal the presence of strong outliers that would necessitate a further reduction of spectral data (see Appendix A). We found the data to be of similar quality across all acquisition parameters and specimens.

### 2.1. Determining Spectral Properties of Necrotic and Vital Tumor Tissue

As described above, we aimed to reduce the heterogeneity of the data set by isolating an identifiable class within all data, i.e., necrosis, which in some cases comprises the entirety of a specimen and can thus be singled out spectroscopically. Our binary classifier showed a solid separability between necrotic and mainly non-necrotic tumor tissue with an overall accuracy of 76% and corresponding AUROC/AUPR values of 0.81/0.81 for the necrosis class and 0.8/0.76 for the vital tumor class (Figure 1A). The misclassification ratio was 0.15 for the necrosis class, indicating a relatively homogenous composition of specimens and 0.31 for the vital tumor class, where mainly (but not exclusively) non-necrotic tissue was expected. Figure 1B displays accompanying performance metrices for each group. Additionally, the classifier performance was evaluated by performing cross validation within the external validation cohort set not only with a patient-wise split but also a random data distribution (see Appendix A). Subsequently, we used feature importance analysis to determine whether individual spectral peaks had a significant part regarding the biochemical composition of the tissue in the classification of groups. However, since the 20 most significant frequency bins only account for 15.16% of the total contribution to the classification—the most significant frequency bins had a significance of only 1.12%—the relevance of the features cannot be limited to a few frequency ranges within our analysis (see Appendix A). It is likely that the heterogenous nature of the specimen analyzed precludes the identification of single molecules or spectral components. The spectral-fingerprint-based binary classifier assigns a probability score to each spectrum that can be used to separate the data set further and perform an additional dimension reduction. By using the calculated optimal threshold of the models internal training validation (0.452, Appendix A), a re-labeling of our entire glioblastoma data set was feasible—all data points (spectra) were classified as necrotic/non-necrotic according to their calculated spectral fingerprint. All data points labeled as vital tissue were placed into a separate class (‘*spectral vital data set*’) with a total of 1310 data points (see Figure 1C).

### 2.2. Spectral Heterogeneity in Vital Glioblastoma

Next, the spectral vital data set was subjected to a dimensional reduction and data display (UMAP). Based on this, we controlled the data for technical confounders such as patient identity and exposure time and did not find any relevant interference (Appendix A). We also plotted points with assumed biological differences (infiltration zone and hemorrhage/blood) mentioned in the histopathological report into the UMAP; here, the distribution was uneven in the UMAP, suggesting that different tissue types were found, as a basis for the observed UMAP pattern (see Appendix A). However, these pathological tags were not sufficient to provide a basis for classification due to the high heterogeneity of the samples. In order to identify different spectral subgroups in the data set, we then performed k-means clustering. Since the number of subgroups was unknown, we chose variable K (1–50) and aimed for a distribution where few classes would hold a majority of the spectra, while sufficient classes for rare spectral subgroups and outliers need to be present (Appendix A). This was the case with a total of 21 clusters (Figure 2B), where 7 clusters contained 95.7% of the Raman spectra and could therefore be considered as major clusters. Four clusters contained less than ten spectra each; ten of the clusters contained only one spectrum. It has to be noted that these clusters still need to be individually present in order to allow us to conduct a similarity-based distribution of the major classes.

We represented all 21 clusters in different colors on the previously established UMAP; this approach independently combined the distribution within the k-means clustering results (each cluster is presented as an individual color), with the spatial arrangement reflecting the dimensional reduction of the previous UMAP analysis. The use of different methods serves as a good control against computational artefacts, allows to assess the quality of the classification and aims to maximize generalization. The clusters were separated very well in the UMAP (Figure 2A), indicating a common origin for each spectral cluster within the tumor tissue.

### 2.3. Gray and White Matter Classify as Distinct Major Clusters

In the next step, non-tumorous brain tissue samples were integrated into the previously established k-means clustering method to predict the cluster that most closely represents the spectral properties of gray and white matter. This analysis resulted in a definite assignment (Figure 2F), since all Raman spectra of the white matter could be classified as cluster 1 (100%), while 47 of the 55 spectra of the gray matter classified as cluster 2 (85%), 6 gray matter spectra were also classified as cluster 1 (11%) and 2 gray matter spectra were classified as cluster 9 (4%). Furthermore, the distribution of clusters 1 and 2 in the UMAP resembles the tissue tagged as an infiltration zone in the histopathological description (Figure 2C) and the spectra with the lowest probability score for being necrotic (Figure 2E). A total of 114 measurements from 181 measurements (63%) diagnosed as infiltration zone indicate an overlap of spectral features of infiltrated brain tissue with clusters 1 (89 spectra) and 2 (25 spectra), as can be seen in Figure 2D.

Here, we show that individually acquired spectra can be identified by utilizing similarity matching to assign them to one of the major clusters in our data set, where the two clusters resemble gray and white matter. While the remaining clusters still need to be characterized, these results can be directly applied to newly generated data.

## 3. Discussion

In this study, we propose a computational and histology-independent way to approach heterogenous spectral features from a highly diverse sample set by dividing Raman measurements of glioblastoma tumor samples into distinct spectral clusters. We separated data emanating from necrotic and vital tissue and identified specific subgroups, which can be verified via independent analysis and by comparing separate tissue samples. While the latter was only feasible for two sets, namely gray and white matter, within the scope of this study, breaking down the spectral heterogeneity from measurements of resected tissue paves the way for using Raman spectroscopy intraoperatively for the identification of tumor boundaries and as a rapid and easy-to-use tool for resection control.

Numerous studies have addressed the capability of tissue identification by means of Raman spectroscopy [12,13,14,15]. These range from the basic task of identifying different anatomical areas of the brain to clinical diagnostic applications in the form of binary (between tumor tissue and healthy brain tissue) and multiclass (to determine distinct tumor entities) classifications [16,17,18,19]. Previously, studies have already dealt with the heterogeneity of glioblastomas. By using frozen tissue samples, Koljenović et al. demonstrated the capability of Raman spectroscopy to distinguish between vital glioblastoma tissue and tumor necrosis, whereas Kalkanis et al. employed Raman spectroscopy to distinguish between healthy brain tissues, vital tumor tissues and necrotic tumors [20]. 

To date, discrimination between tumors and surrounding tissues typically relies on the correlation of measurements with histopathology. This correlation can be augmented by measuring tissue fragments as small as possible to minimize histomorphological complexity [21]; however, in glioblastoma, single spatial entanglements of characteristic pathological features often occur even within small tumor samples. When adding the aspect of each sample’s three-dimensionality, the section used for the pathologic assessment in comparison to the spectroscopic measurement on the surface of the sample increases the rate of miscorrelation. The use of formalin-fixed paraffin-embedded (FFPE) tissue offers the benefit of obtaining corresponding measurements with regard to distinct areas identified by means of light microscopy. In this sense, we were recently able to spectroscopically assess the histomorphological heterogeneity of glioblastoma and classify peritumoral tissue, tumor tissue and necrosis based solely on spectroscopic FFPE tissue properties [22]. The group of Amharref et al. further demonstrated the representation of different biochemical properties of tumor necrosis in the form of different spectroscopic characteristics and identified central necrotic areas with a high protein content and a peripheral area with an increased lipid content [23]. However, such types of tissue processing severely alter the sample’s biochemistry and are not directly applicable during surgery; therefore, native tissue was used in this study. Our in silico approach overcomes these limits of classical application of spectroscopy regarding tumor samples. While necrotic samples were initially identified with histopathology, the remaining classification was undertaken by k-means cluster analysis; a number of 21 clusters could be identified in the patient data. Fourteen of these clusters contained less than ten spectra each. We suppose that these clusters represent distinct but rare entities, such as bone chips or burnt tissue. Notably, this approach requires no curation of data as spectral outliers will constitute distinct clusters, rendering it particularly useful in a real-world scenario. The main bulk of spectral data that are represented in seven clusters can be presumed to arise from the underlying vital tumor tissue as well as from the infiltrated and peritumoral gray and white matter. We show a notable correlation of clusters 1 and 2 with the histopathologically diagnosed infiltration zone, which refers to the areas that are not mainly constituted by the tumor itself, be it vital or necrotic, but by brain tissue with malignant cells in variable numbers. It is of particular importance to identify these transition zones during surgery, and the presented identification of these clusters may serve to do so. Moreover, clusters 1 and 2 were also determined by our established predictive model to be the least likely to be necrotic. The necrosis-classifier-based UMAP and the k-means clustering are completely independent methods that arrive at the same conclusion, strengthening reliability. The clear assignment of the measurements of the healthy control brain tissue into only these two clusters confirms this and further identifies the spectral origin as infiltrated and peritumoral gray and white matter. Moreover, our data suggest that there is an accumulation of a wider range of spectroscopic features of the infiltration zone rather than a single defining spectrum, similar to the known spectroscopic heterogeneity of tumor necrosis [23]. In the case of peritumoral brain tissue and infiltration zones, this may be related to a differing amount of tumor infiltrating cells, a diverging amount of associated reactive gliosis and different anatomical areas of infiltration as well as an associated metabolic shift. In the next step, a precise assignment of the remaining clusters to specific tumor areas would be useful, although this would require even more precise pathological matching with the tumor samples or further homogenous samples of tumor tissue analog to the gray and white matter approach. This may be achieved by measuring cultured tumor cells in vitro but is beyond the scope of this computational approach.

Our study reflects the actual conditions in the operating room. To subject patient-specific healthy tissue to spectroscopic examination, a portable, handheld Raman tool [24] could be employed intraoperatively. Within this approach, even in vivo measurements of tumor borders and/or resection margins seem feasible [11]. However, a direct application in vivo would entail the operation having to be prolonged and the acceptance of possible complications, which have hampered the application of this approach. In order to justify the additional burden of a routine use of an in vivo application, sufficient evidence of its feasibility needs to be presented; we believe that our study delivers this to the scientific and medical community. In the subsequent second step, spectroscopy could then be evaluated by several research groups as a potential technique for in vivo diagnostics. Our study resolves the problem of large heterogeneity of spectra acquired from glioblastoma samples, which has limited the usefulness of this approach so far. Our comparison with healthy autopsy tissue could serve as the proof of plausibility for the application of spectroscopy to determine resection margins and infiltration zones. 

## 4. Materials and Methods

### 4.1. Patient Data

Presented patient data were collected between 2018 and 2021 (INSITU study—Intraoperative Spectroscopy and Imaging Tumors—approved by the ‘Comité National d’Ethique de Recherche’; CNER—No. 201804/08). Experiments were conducted in accordance with the ‘EU General Data Protection Regulation’ (GDPR) [25] and the ‘Declaration of Helsinki of the World Medical Association’ [26]. Pre-experimentally, all patients were informed about the study design and each patient provided written consent to participate in this study. In total, 1456 intraoperative tumor measurements from 43 glioblastoma tumor cases as well as 87 measurements of gray and white matter with absent pathological findings from autoptic brain were examined (Table 1). All tumor samples underwent a perioperative spectroscopical examination directly after the surgical resection/biopsy was carried out by a board-certified neurosurgeon at the Centre Hospitalier de Luxembourg (CHL, Luxembourg). The autopsy of the deceased patient was performed at the Laboratoire nationale de Santé (LNS, Luxembourg); obtained brain tissue samples were subsequently measured as healthy control at the CHL 16 h post-mortem. Histological tissue diagnostic was performed by a board-certified neuropathologist at the National Center of Pathology (NCP) at the LNS; additional techniques, such as immunohistochemistry, and (epi-)genetic analyses were carried out for the purpose of integrated diagnosis according to the fifth edition of the WHO Classification of Tumors of the Central Nervous System [27].

### 4.2. Tissue Preparation and Data Acquisition

All Raman measurements of glioblastoma tumor samples were carried out perioperatively; by placing the Raman spectrometer in close proximity to the surgical procedure, rapid data acquisition of fresh tissue samples was feasible. After tumor samples were collected (tumor excision was performed), they were hydrated in physiological saline—a standard procedure that prevents dehydration or further destruction of the tissue and its biochemical composition, and that is also mirrored by the operation situs, which is washed and moisturized with the same solution. For data acquisition, a robotic visualization and spectroscopic acquisition system was used; the collection of Raman spots was achieved by placing the tumor excisates in the focal point of a Raman spectrometer (Solais™, Synaptive^®^, Toronto, ON, Canada) with a motorized stage. Within each tumor sample, up to 25 measurement points (id est, spots of interest) were determined and measured with 1 to 30 acquisitions to reduce random variations (noise) in individual measurements and to boost the signal-to-noise ratio. All Raman spectra were acquired using a 785 nm laser (output power 50 mW, maximum penetration depth 1 mm, 80 μm diameter of the laser spot) with an acquisition time of 0.7 to 10 s. The parameter variation served as an assessment of the robustness of this method, and we did not find bias in the analysis (see Appendix A). By aiming to achieve an optimal correlation between collected Raman spectra and the subsequent histopathological diagnosis, all examined specimens were constrained to an approximate size of 5 mm maximum. During the spectroscopic examination, biological samples were placed in an aluminum cup. Due to its negligible spectral background contribution, aluminum serves as a favorable and low-cost Raman substrate [28], especially during the measurement of small tissues. After data acquisition was completed, all tumor samples were placed in a formalin solution (4%) and underwent neuropathological diagnosis, viz. light microscopic as well as (epi)-genetic and immunohistochemical examination; see Figure 3 for an overview of the study design. During the pathological examination, each tumor fragment was individually microscopically examined and analyzed; this description allowed us to assign morphological features (such as necrosis, infiltration zones, hemorrhage) to individual tumor specimens.

### 4.3. Data Analysis Sequence and Machine Learning

In order to determine the different classes of spectra emerging from measurements on glioblastoma fragments, we applied a two-step approach where, first, a putative spectrum representing necrosis was identified and all data were classified and visualized in relation to this; in the second step, the data were clustered in an independent manner and mapped the results of the first step. Raman data were initially split into three groups (id est, classes) based on the description of the samples’ morphology in the pathological report and the underlying histopathological features. The first class (‘*necrosis data set*’) included Raman measurements of tumor specimens consisting mainly (more than 80%) of necrosis. Histopathologically, the *necrosis class* consists of unstained and avital nuclei as well as fragments of apoptotic nuclei in an eosinophilic surrounding. The second class (*‘vital data set’*) included Raman measurements of tumor areas with either densely packed vital tumor cells showing pleomorphic nuclei and an increased mitotic rate or infiltrative tumor areas (infiltration zones, histomorphologically consisting of migrating tumor cells and a moderate to low increase in cell density) and peritumoral zone (transition from infiltration zone to non-pathological gray and white matter). The third class *(‘heterogenous data set’*) contains spectral data from specimens of the same patients where no information about the specific proportion of the respective tissue types was provided. Therefore, all characteristic features of glioblastoma, such as necrosis and vital tumor areas, as well as infiltrative tumor areas and peritumoral zones, can be found in the *heterogenous class*. Table 2 provides an overview about the number of spectra/patients in each of the three classes.

Afterward, a baseline correction and fluorescent signal removal were completed with the software (Solais, Synaptive, Version 1.0) directly on the instrument using a Savitzky–Golay filter, individual recordings were cleaned and cosmic ray artefacts were removed from the spectra [29]; several measurements were labeled as outliers based on the visible shape of the spectrum (e.g., hot pixels, oversaturation). As we could not initially determine their origin, we identified these spectra in a subsequent analysis (distance-based hierarchical dendrogram) to control for potential cofounding effects. After standardization via a spectrum and frequency bin, agglomerative clustering was performed, and the top three levels were plotted; they did not indicate the presence of strong outliers, which would consecutively require additional trend/outlier removal. For the initial data overview, all established three data classes (‘*necrosis data set*’, ‘*vital data set*’, *heterogenous data set*’) and autopsy brain tissue, which served as a healthy control, were visualized (mean spectra and variance) (see Appendix A). 

To initially determine unique spectroscopic properties of necrotic and non-necrotic glioblastoma tissue, we performed a binary supervised learning classification using random forest analysis (algorithm), including hyperparameter tuning with internal cross-validation (internally, 5 splits were conducted and repeated 3 times). Classifier performance was evaluated using AUROC/AUPR, macro and weighted average, and precision, recall and f-1 score. To reduce the potential patient-specific classification and data imbalances, data were split independently (per patient) in a training and validation cohort, and the number of measurements per patient was equally set to 15 (see Table 3). The optimal decision threshold of the internal cross-validation was calculated based on the f-1 score and used in the next step to evaluate the spectral tissue features in our data sets. All glioblastoma measurement points were re-curated based on their spectral properties—a data set consisting solely of non-necrotic tissue according to its spectroscopic properties was established (‘*spectral vital data set*’).

Afterward, we employed the dimension reduction technique UMAP (Uniform Manifold Approximation and Projection) on the ‘*spectral vital data set*’ to diminish high-dimensional spectral data into a lower-dimensional space, while maintaining local and global structural integrity of the data. Hereby, Raman spectra were reduced to n = 30 (n = 15 in the ‘*vital data set*’, respectively) per patient to reduce potential patient-specific spectra bias. Furthermore, the optimal decision threshold (0.452) for a distinction between ‘*necrosis data set*’ and ‘*vital data set*’ was used to re-evaluate the spectral properties of all data points, and the probability of them being computationally classified as spectroscopically necrotic was displayed. Additionally, we used the vector quantization technique, k-means clustering, to determine the WSS (within-cluster sum of squared errors) score for kmax = 50 (hereby, the Euclidian distance served as a distance metric) on the ‘*spectral vital data set*’; moreover, we searched for the most reasonable threshold in the sum of squared errors within centroids with increasing k clusters [30]. Both aforementioned techniques serve as computational approach to assess and visualize the spectral heterogeneity within vital glioblastoma tissue in an unsupervised way. Findings were evaluated using the histopathological report and non-pathological brain tissue as the healthy control. Technical details about our computational data approach are described in the Appendix A.

## Figures and Tables

**Figure 1 molecules-29-00979-f001:**
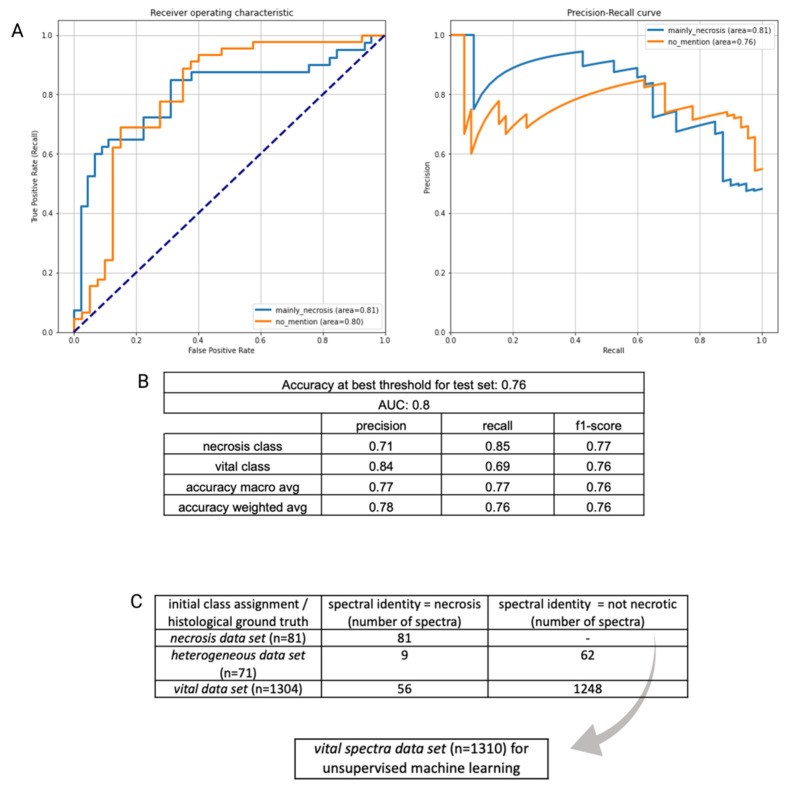
(**A**): ROC and PR curve of initial binary classification (*vital class/necrosis class*). (**B**): Corresponding performance metrics. (**C**): The optimal decision threshold of the binary classification was used to re-evaluate and re-assign spectra to a certain class according to their spectroscopic behavior. A *spectral vital data set* (gray arrow), in which spectral properties of necrosis were excluded, was used in the next step to determine spectral heterogeneity within vital tumor tissue.

**Figure 2 molecules-29-00979-f002:**
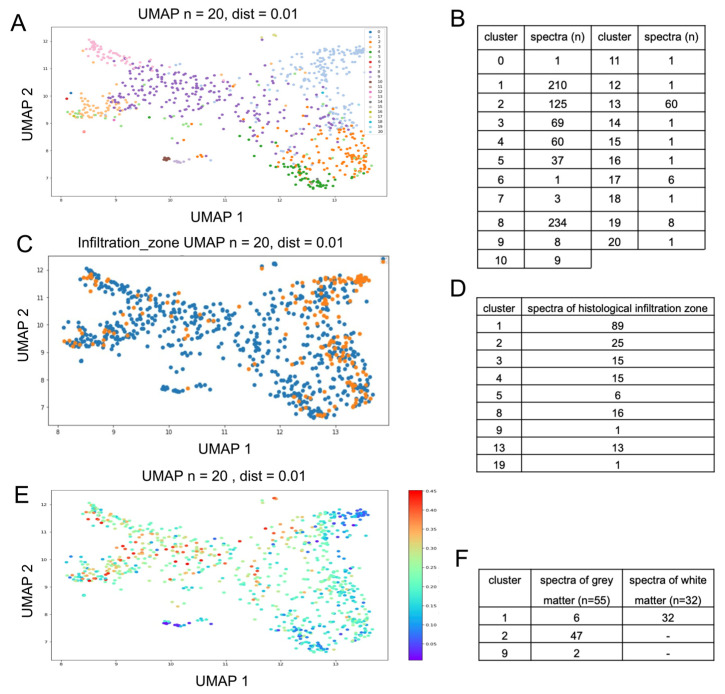
(**A**): Twenty-one clusters from k-means clustering, each in different colors, are represented using UMAP. (**B**): Data distribution within the 21 clusters. (**C**,**D**): UMAP; histomorphologically determined infiltration zones are represented in orange (**C**). An overlap of spectral features of infiltrated brain tissue with clusters 1 and 2 can be determined ((**D**) shows total number of spectra identified as infiltration zone according to their spectroscopic cluster assignment). (**E**): UMAP representation with overlaid predicted probability of necrosis. (**F**): assignment of healthy brain tissue to distinct spectral clusters.

**Figure 3 molecules-29-00979-f003:**
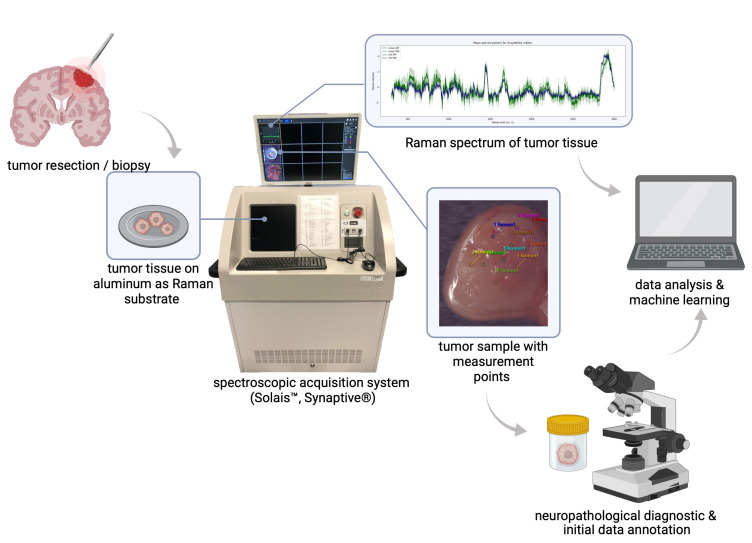
Overview of study design and workflow. After tumor resection, surgical specimens were spectroscopically examined. In accordance with the neuropathological diagnostic and histomorphological hallmarks, inherent spectral properties of a single tumor type can be used for a machine learning-based classification of tumor heterogeneity.

**Table 1 molecules-29-00979-t001:** Overview of patient data and spectroscopic measurements.

	Glioblastoma	Autoptic Brain Tissue
number of tumor cases	43	1
number of measurements	1456	87

**Table 2 molecules-29-00979-t002:** Overview of Raman measurements in each class of the initial classification.

Initial Class Assignment/ Histological Ground Truth	*Necrosis Data Set*	*Vital Data Set*	*Heterogeneous Data Set*
number of Raman measurements (*n*)	81	1304	71

**Table 3 molecules-29-00979-t003:** Data split and patient distribution in the initial binary classification approach.

	*Necrosis Data Set (n = 81)*	*Vital Data Set (n = 136)*
number of measurements in training set (8 patients)	41	91
number of measurements in external validation set (3 patients)	40	45

## Data Availability

Interested parties are asked to contact the corresponding author for personalized options.

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
