# Peer review of "Computational Assessment of Spectral Heterogeneity within Fresh Glioblastoma Tissue Using Raman Spectroscopy and Machine Learning Algorithms"

_molecules, 2024, doi:10.3390/molecules29050979_

Round 1

Reviewer 1 Report

Comments and Suggestions for Authors

In this manuscript, the authors described “Computational Assessment of Spectral Heterogeneity within Fresh Glioblastoma Tissue Using Raman Spectroscopy and Machine Learning Algorithms”. However, there are many problems in experimental results, especially in the interpretation of the results. Thus, I would recommend this paper should be rejected.

Introduction:

1. The current research background is not comprehensive enough, as it does not overview the current diagnostic methods for brain tumors and their limitations, nor does it describe the current applications of Raman spectroscopy in diagnosing brain tumors.

2. A brief overview of the algorithms and methods involved in the methodology of this study is lacking.

Materials and Methods:

1. The patient's age, gender and other information were not provided, which may affect the experimental results.

2. How to generalize each class?

Results:

1. The explanation of the results lacks sufficient depth, especially under the circumstance where the Raman intensity is so weak. How can the credibility of the results be ensured?

2. The paper does not fully discuss the experimental results and lacks analysis and discussion of the original Raman spectra. What are the spectral differences between three classes and whether they reflect different component information? Are there any common spectral features within the spectral cluster and What are the differences between the spectral clusters after K-means clustering.

3. The paper does not explain how single-point Raman measurement realizes tumor edge definition.

4. How to evaluate the model performance of K-means clustering? What are the advantages of using this technology for tumor edge identification compared to other spectral imaging identification methods?

5. Spectral preprocessing has a significant impact on the results of the model, the paper does not explain the preprocessing used by the model.

6. The paper does not provide necessary introduction of the algorithms used, such as K-means clustering, and UMAP.

Discussion

1. Discussion should be expanded comparing your results with the existed literature.

Comments on the Quality of English Language

Minor editing of English language required

Author Response

Point by point response to reviewer#1.

Title: Computational Assessment of Spectral Heterogeneity within Fresh Glioblastoma Tissue Using Raman Spectroscopy and Machine Learning Algorithms

Manuscript-ID: molecules-2743982

Corresponding author: Dr. Klamminger and Dr. Kleine Borgmann, gilbert-georg.klamminger@uks.eu

We thank the reviewer for the thorough interrogation of our work. Please find below a detailed point-by-point response. We hope that they will find this work worthy of publication based on the additional analysis and changes and improvements made to the manuscript as well as our discussion.

REVIEWER:

Introduction:

  1. The current research background is not comprehensive enough, as it does not overview the current diagnostic methods for brain tumors and their limitations, nor does it describe the current applications of Raman spectroscopy in diagnosing brain tumors.
  2. A brief overview of the algorithms and methods involved in the methodology of this study is lacking.

ANSWER:

We thank the reviewer for the comments and input and adapted our introduction accordingly.

As requested, we added a brief overview of both the machine learning algorithms and spectroscopical methods involved in the methodology, see page 2.

Additionally, we highlight not only details of current applications of Raman spectroscopy in neuro-oncology (spectroscopical detection of tumorous tissue, as well as of different types and grades of brain tumors) but also name the diagnostic modalities of brain tumor assessment to date to draw a comprehensive research background for readers from different subspecialities, see also page 1 and 2.

REVIEWER:

            Materials and Methods:

  1. The patient's age, gender and other information were not provided, which may affect the experimental results.
  2. How to generalize each class?

ANSWER:

1.) We thank the reviewer for the hint and added the clinical and biological information, see the new Supp. Table 1.

2.) The key to generalization is to identify features that define “natural” (biological) groupings (in our case: distinctly different tumor areas) from a complex spectral data set not only within a certain environment but also when confronted with unseen input. That said, we subsequently validated our supervised learning algorithm using a cross validation within an external validation cohort with a patient-wise split but also a random data distribution and visualized as well compared the results of two independent unsupervised learning techniques (UMAP, k-means clustering) among one another. Together with the control of the data for technical confounders (e.g., patient identity, exposure time, see Supp. Fig. 5) using dimensional reduction in combination with comparison of particular classes to histologically determined gray and white matter (= healthy control), we do not only reduce overfitting but also aim at maximized generalization. See page 4: “The use of different methods serves as a good control against computational artefacts, allows to assess the quality of the classification and aims to maximize generalization.”

REVIEWER:

Results:

  1. The explanation of the results lacks sufficient depth, especially under the circumstance where the Raman intensity is so weak. How can the credibility of the results be ensured?
  2. The paper does not fully discuss the experimental results and lacks analysis and discussion of the original Raman spectra. What are the spectral differences between three classes and whether they reflect different component information? Are there any common spectral features within the spectral cluster and What are the differences between the spectral clusters after K-means clustering.
  3. The paper does not explain how single-point Raman measurement realizes tumor edge definition.
  4. How to evaluate the model performance of K-means clustering? What are the advantages of using this technology for tumor edge identification compared to other spectral imaging identification methods?
  5. Spectral preprocessing has a significant impact on the results of the model, the paper does not explain the preprocessing used by the model.
  6. The paper does not provide necessary introduction of the algorithms used, such as K-means clustering, and UMAP.

ANSWER:

1.) To ensure credibility of the presented results especially under the circumstance of the received Raman intensity several measures of validation have been implemented:

- During sample preparation: Raman spectra were acquired as average measurements (up to 30 averages per spot of interest) - this technique reduces random variation within the data. We added this aspect to our manuscript, see page 8: “Within each tumor up to 25 measurement points (id est spots of interest) were determined and measured with 1 to 30 acquisition to reduce random variations (noise) in individual measurements and boosting the signal-to-noise-ratio.”

- During computational learning and data interpretation: Standardization by spectrum and frequency bin was employed, aiming at maximized consistency of measurements, see also point 5 of this answer below (spectral preprocessing). Furthermore, the overall validity of our computational results is undermined by robust machine learning algorithms, including internal cross-validation and testing with external data set and the basis of our initial basis classification (discrimination between vital tissue und necrosis) is based on the gold standard histopathology.

2.) The reviewer points out an existential question for the use of Raman spectroscopy as a diagnostic tool. While some research groups strive to determine certain peaks of interests (proteins/lipids) to analyze the underlying specimen1,2, we refrain from interpreting Raman bands and define a spectrum as a diagnostic “fingerprint” for classification (diagnostic approach). In a different context, we used feature discrimination to distinguish dura mater and meningioma3, where we found that few features of the spectrum representing collagen were sufficient to perform classification.

We also explored the classifications in the case of highly heterogenous glioma and brain spectra with a feature importance analysis to determine if individual spectral peaks had a significant part regarding the biochemical composition of the tissue. However, since the 20 most significant frequency bins account for only 15.16% of the total contribution to the classification - the most significant frequency bins had a significance of just 1.12% - the relevance of the features cannot be limited to a few frequency ranges within our analysis and a more detailed examination of these features not useful. We added this point and the underlying feature analysis to our manuscript, see “Results” section on page 3 and the new Suppl. Figure 3.

3.) In contrast to the acquisition of spectroscopic maps, we employ a single spot-measurement-based method. Within the scope of our study this point-mapping approach provides a fast readout, which is largely insensitive to sampling errors4 and is associated with the possibility of longer acquisition times and superior spectra quality5. The spot size is chosen to contain a representation of the local tissue composition to allow for meaningful single measurements. Regarding the spectroscopic properties of tumor infiltration zone, the reviewer points out an important aspect of the application of RS in neurooncological surgery, where the infiltration zone is of particular interest to determine the extent of resection necessary:  The vast amount of data is represented in 7 clusters - cluster 1 and 2 do not only resemble tissue samples tagged as infiltration zone in the histopathological description (see Figure 2C) but share spectroscopic features with grey and white matter (see Figure 2F).

That said, our data suggests the accumulation of a broader range of ‘spectroscopic infiltration zone features’ rather than a single defining spectrum, similar to spectroscopic features of tumor necrosis6. This may be related to a differing amount of tumor infiltrating cells, diverging amount if associated reactive gliosis and different anatomical areas of infiltration. We hope that our findings, which can be directly applied to newly generated data, contribute significantly to the applicability in a subsequent in vivo approach.

We added this important point to our manuscript, see page 7: “The clear assignment of the measurements of the healthy control brain tissue into only these two clusters confirms this and further identifies the spectral origin as infiltrated and peritumoral gray and white matter. That said, our data suggest the accumulation of a wider range of spectroscopic features of the infiltration zone rather than a single defining spectrum, similar to the known spectroscopic heterogeneity of tumor necrosis [25]. In the case of peritumoral brain tissue and infiltration zones, this may be related to a differing amount of tumor infiltrating cells, a diverging amount of associated reactive gliosis and different anatomical areas of infiltration.

4.) We thank the reviewer for bringing to our attention that this point needs to be clarified to the readership, as it is a central part of the identification process. The advantage of K-means- clustering with variable K is that a completely unbiased sorting of spectra is undertaken without the need of supervision by the examiner. This is paramount if a future in-situ use is anticipated, as visual control of a spectrum would hamper the process. At the same time, it allows for an unlimited number of different classes to be included in the classification algorithm and not every source of artifactual spectral sources can be anticipated beforehand (e.g. traces of material from the surgery).

The performance is evaluated by a separate and independent method, here UMAP dimension reduction, which is plotted along with the result of the k-means-clustering (Figure 3). The separation of clusters within the UMAP strongly suggests a high performance of the k-means-clustering (also see discussion, line 299).

5.) We thank the reviewer for bringing up this issue and therefore added the necessary details about preprocessing within the main text of our manuscript, see page 9: “After baseline correction and fluorescent signal removal are done by the software on the commercial instrument using a Savitzky-Golay filter, individual recordings were cleaned, and Cosmic ray artefacts were removed from the spectra; several measurements were labeled as outliers based on the visible shape of the spectrum (e.g., hot pixels, oversaturation). As we could not initially determine their origin, we identified those spectra in subsequent analysis (distance-based hierarchical dendrogram) to control for potential cofounding effects. After standardization by spectrum and frequency bin, agglomerative clustering was performed, and top three levels were plotted; they did not indicate the presence of strong outliers which consecutively would require additional trend/outlier removal. For initial data overview all established three data classes ((‘necrosis data set’, ‘vital data set’, heterogenous data set’) and autopsy brain tissue, which served as healthy control, were visualized (mean spectra and variance), see Supp. Fig. 6.” 

6.) To provide the reader detailed information of the algorithms used we added extensive bioinformatical details of the employed learning strategies within the Material and Method Section, see page 9. Additionally, see also page 2 for an introductive overview of the machine learning algorithms involved in our study (see question 2, introduction above).

REVIEWER:

Discussion: Discussion should be expanded comparing your results with the existed literature.

ANSWER:

As requested, we expanded our discussion and discussed the relevant literature, see page 6.

REVIEWER:

            Comments on the Quality of English Language:

Minor editing of English language required

ANSWER:

Grammar and spelling mistakes as well as potentially misleading phrases / wordings were discussed within the team and corrected where necessary.

Literature:

1Sivasubramanian Murugappan, Syed A. M. Tofail, and Nanasaheb D. Thorat. Raman Spectroscopy: A Tool for Molecular Fingerprinting of Brain Cancer. ACS Omega 2023, 8 (31), 27845-27861, doi: 10.1021/acsomega.3c01848

2Payne, T.D.; Moody, A.S.; Wood, A.L.; Pimiento, P.A.; Elliott, J.C.; Sharma, B. Raman Spectroscopy and Neuroscience: From Fundamental Understanding to Disease Diagnostics and Imaging. Analyst 2020, 145, 3461–3480, doi:10.1039/D0AN00083C.

3Jelke, F., Mirizzi, G., Borgmann, F.K. et al. Intraoperative discrimination of native meningioma and dura mater by Raman spectroscopy. Sci Rep 11, 23583 (2021). https://doi.org/10.1038/s41598-021-02977-7

4Klamminger GG, Gérardy J-J, Jelke F, et al. Application of Raman spectroscopy for detection of histologically distinct areas in formalin-fixed paraffin-embedded glioblastoma. Neuro-Oncology Adv. 2021;3(1):vdab077. doi:10.1093/noajnl/vdab077

5Butler HJ, Ashton L, Bird B, et al. Using Raman spectroscopy to characterize biological materials. Nat Protoc. 2016;11(4):664-687. doi:10.1038/nprot.2016.036

6Amharref, N.; Beljebbar, A.; Dukic, S.; Venteo, L.; Schneider, L.; Pluot, M.; Manfait, M. Discriminating Healthy from Tumor and Necrosis Tissue in Rat Brain Tissue Samples by Raman Spectral Imaging. Biochim Biophys Acta Biomembr 2007, 1768, 2605–2615, doi:10.1016/j.bbamem.2007.06.032.

Reviewer 2 Report

Comments and Suggestions for Authors

Overall, I think the manuscript has some merit and can be published.

It is batter to clearly indicate how sample is the tissue samples are prepared and RS data are collected. Is the tissue samples sliced or ? As I understand,  you did not of the tissue as you indicate  1 to 30 acquisitions from per sample is performed. I believe spectral acquisition less than 20 from per sample for this type of data processing is not satisfactory. Please clarify this point.

What do you think about the laser penetration depth into tissue? This point is important as you study the heterogeneity of the tissues. 

Changing the spectral acquisition time is not a good idea for this type of study since it will effect the obtained spectral information causing deviations from the goal of the study.

Where is the laser power 50 mW, on the sample or at the laser exit? Please indicate this in the manuscript. 

Comments on the Quality of English Language

A few mistakes here or there. 

Author Response

Point by point response to reviewer#2.

Title: Computational Assessment of Spectral Heterogeneity within Fresh Glioblastoma Tissue Using Raman Spectroscopy and Machine Learning Algorithms

Manuscript-ID: molecules-2743982

Corresponding author: Dr. Klamminger and Dr. Kleine Borgmann, gilbert-georg.klamminger@uks.eu

REVIEWER:

Overall, I think the manuscript has some merit and can be published.

ANSWER:

We thank the reviewer for his work and the positive assessment of our work.

Please find below our detailed point-by-point response to each comment. Furthermore, we integrated all points raised within our revised manuscript. 

REVIEWER:

It is batter to clearly indicate how sample is the tissue samples are prepared and RS data are collected. Is the tissue samples sliced or ? As I understand,  you did not of the tissue as you indicate  1 to 30 acquisitions from per sample is performed. I believe spectral acquisition less than 20 from per sample for this type of data processing is not satisfactory. Please clarify this point.

ANSWER:

We thank the reviewer for the comment:

Since the interested reader may point out the same question regarding the placement/slicing of samples, we added the missing information also within our manuscript as follows, see page 8:

“All specimens were constrained to an approximate size of 5 mm maximum, aiming at optimal correlation between collected Raman spectra and the subsequent histopathological diagnosis.”

Tissue was used as extracted in surgery without further processing or slicing (see Materials). Up to 25points of interest (= spots of interest within the tumor tissue) were determined on individual tumor samples.   Hereby each selected point of interest was measured with 1 to 30 acquisition averages, an efficient way to reduce random variations (“noise”) in individual measurements and leveraging the signal-to-noise-ratio.

As correctly pointed out by the reviewer there must be a sufficient number of selected spots per sample - therefore, up to 25 different points of interest (= spots of interest within the tumor tissue) were selected within our study.

We added the missing information within our manuscript, see page 8: “Within each tumor sample, up to 25 measurement points (id est spots of interest) were determined and measured with 1 to 30 acquisition to reduce random variations (noise) in individual measurements and boosting the signal-to-noise-ratio.” The variation in acquisition parameters served as a control for cofounding variables (acquisition time and averaging for noise suppression), as discussed in the manuscript. We agree with the reviewer that it is paramount to obtain a sufficient amount of data to perform such a study. Here, it is important to note that several samples are measured for each case, each sample is analyzed at up to 25 spots (depending on the size) and each spot is measured repeatedly to account for measurement artifacts. For more details, please refer to the supplemental material. 1543 spectra of 44 individuals (35 spectra per n on average) were generated on this way by taking approximately 25000 individual measurements.

REVIEWER:

What do you think about the laser penetration depth into tissue? This point is important as you study the heterogeneity of the tissues.

ANSWER:

We thank the reviewer for this enriching comment and would like to highlight this important technical issue: 

Although certain modifications of the Raman effect (such as Tip-enhanced Raman Spectroscopy - TERS) can be used to require a high three-dimensional resolution of the specimen of interest, our method of spontaneous Raman scattering using a single point resolution with automated integrated focus setting allows for intermitted analysis of the tissue heterogeneity of the respective analyzed surface, with a maximum penetration depth of 1mm1. It should be noted that the penetration depth strongly depends on the opacity and absorption of the tissue for the used wavelength. We conducted a series of tests in this regard and found that at more than 1mm sample thickness, no signal from material placed underneath could be detected.

We added this technical information, together with the diameter of the spot size, to our manuscript, seepage 8:

Raman spectra were acquired using a 785nm laser (output power 50mW, maximum penetration depth 1mm, 80μm diameter of the laser spot) with an acquisition time of 0.7 to 10 sec.”

REVIEWER:

Changing the spectral acquisition time is not a good idea for this type of study since it will effect the obtained spectral information causing deviations from the goal of the study.

ANSWER:

For clinical use, a method would need to be robust enough to be largely parameter-independent. We included parameter-variation for acquisition time and noise-suppressing averaging to test this robustness and found neither of these to be a factor in data quality or a cofounding variable in the classification process (see supplemental methods and discussion, assessment of the heterogenous data distribution within the corresponding UMAP of this feature, no relevant bias due to different acquisition parameters was found). We added this information on page 2 and 8.

REVIEWER:

Where is the laser power 50 mW, on the sample or at the laser exit? Please indicate this in the manuscript.

ANSWER:

The mentioned laser power (50 mW) is defined as output power. We clarified this also within our manuscript, see page 8.

REVIEWER:

Comments on the Quality of English Language: A few mistakes here or there.

ANSWER:

Grammar and spelling mistakes as well as potentially misleading phrases / wordings were discussed within the team and corrected where necessary.

Literature:

  • Mensah-Brown, Derek Yecies, Gerald A Grant, Rapid intraoperative diagnosis of pediatric brain tumors using Raman spectroscopy: A machine learning approach, Neuro-Oncology Advances, Volume 4, Issue 1, January-December 2022, vdac118, https://doi.org/10.1093/noajnl/vdac118

Round 2

Reviewer 1 Report

Comments and Suggestions for Authors

The spectral response values are too low to distinguish whether the difference is caused by noise or the sample difference. So, the article has serious flaws.Thus, I would recommend this paper should be rejected. 

Comments on the Quality of English Language

 Minor editing of English language required.

Author Response

Point by point response to reviewer#1. - round 2.

Title: Computational Assessment of Spectral Heterogeneity within Fresh Glioblastoma Tissue Using Raman Spectroscopy and Machine Learning Algorithms

Manuscript-ID: molecules-2743982

Corresponding author: Dr. Klamminger and Dr. Kleine Borgmann, gilbert-georg.klamminger@uks.eu

REVIEWER:

The spectral response values are too low to distinguish whether the difference is caused by noise or the sample difference. So, the article has serious flaws. Thus, I would recommend this paper should be rejected.

ANSWER:

We thank the reviewer for the fruitful discussion of our manuscript and believe that their input has allowed us to make our findings more poignant to the readership, please refer to the first point-by-point response and the revised version of the manuscript.

We are convinced that our data, which is well in line with the literature and shows a convincing pattern in classification, merits publication and will be a contribution to the field that will be built upon.

Reviewer 2 Report

Comments and Suggestions for Authors

The authors addressed to my concerns but interestingly they have now realized the importance of those mentioned technical points. This indeed does not look good for the accuracy of the reported data. It is still not clear to me how Raman spectra from each sample is collected. 

Comments on the Quality of English Language

It reads enough.

Author Response

Point by point response to reviewer#2.

Title: Computational Assessment of Spectral Heterogeneity within Fresh Glioblastoma Tissue Using Raman Spectroscopy and Machine Learning Algorithms

Manuscript-ID: molecules-2743982

Corresponding author: Dr. Klamminger and Dr. Kleine Borgmann, gilbert-georg.klamminger@uks.eu

REVIEWER:

The authors addressed to my concerns but interestingly they have now realized the importance of those mentioned technical points. This indeed does not look good for the accuracy of the reported data. It is still not clear to me how Raman spectra from each sample is collected.

ANSWER:

We thank the reviewer for the thorough discussion and agree that the assessment of the technical robustness was an interesting and important addition.

The collection of the Raman spots was performed by placing fresh and untreated Tumor excisates in the focal point of a Raman spectrometer with a motorized stage (Synaptive Solais). We revised the manuscript according to the referee’s comment to make this more clear; please see page 8 for the adapted and detailed description of our workflow: “After tumor samples were collected (tumor excision was performed), they were hydrated in physiological saline - a standard procedure that prevents dehydration or further destruction of the tissue and its biochemical composition and that is also mirrored by the operation situs, which is washed and moisturized with the same solution. For data acquisition a robotic visualization and spectroscopic acquisition system was used; the collection of Raman spots was achieved by placing the tumor excisates in the focal point of a Raman spectrometer (Solais™, Synaptive®, Toronto, Canada) with a motorized stage. Within each tumor sample, up to 25 measurement points (id est spots of interest) were determined and measured with 1 to 30 acquisitions to reduce random variations (noise) in individual measurements and boosting the signal-to-noise-ratio. All Raman spectra were acquired using a 785nm laser (output power 50mW, maximum penetration depth 1mm, 80μm diameter of the laser spot) with an acquisition time of 0.7 to 10 sec. The parameter variation served as an assessment of the robustness of this method, we did not find a bias in the analysis (see Supp. Fig. 5). Aiming at optimal correlation between collected Raman spectra and the subsequent histopathological diagnosis, all examined specimens were constrained to an approximate size of 5 mm maximum. During spectroscopic examination, biological samples were placed in an aluminum cup. Due to its negligible spectral background contribution, aluminum serves as a favorable and low-cost Raman substrate [28], especially during measurement of small tissues. After data acquisition was completed, all tumor samples were placed in formalin solution (4%) and underwent neuro-pathological diagnosis, viz. light microscopic as well as (epi)-genetic and immuno-histochemical examination; see Fig. 3 for an overview of the study design.”.
